# P4GCN: Vertical Federated Social Recommendation with Privacy-Preserving Two-Party Graph Convolution Networks

## ABSTRACT

In recent years, graph neural networks (GNNs) have been commonly utilized for social recommendation systems. However, real-world scenarios often present challenges related to user privacy and business constraints, inhibiting direct access to valuable social information from other platforms. While many existing methods have tackled matrix factorization-based social recommendations without direct social data access, developing GNN-based federated social recommendation models under similar conditions remains largely unexplored. To address this issue, we propose a novel vertical federated social recommendation method leveraging privacy-preserving two-party graph convolution networks (P4GCN) to enhance recommendation accuracy without requiring direct access to sensitive social information. First, we introduce a Sandwich-Encryption module to ensure comprehensive data privacy during the collaborative computing process. Second, we provide a thorough theoretical analysis of the privacy guarantees, considering the participation of both curious and honest parties. Extensive experiments on four real-world datasets demonstrate that P4GCN outperforms state-of-the-art methods in terms of recommendation accuracy.

## KEYWORDS

Social Recommendation, Federated Learning, Graph Neuron Network

**ACM Reference Format:**
Anonymous Author(s). 2024. P4GCN: Vertical Federated Social Recommendation with Privacy-Preserving Two-Party Graph Convolution Networks. In *Proceedings of Make sure to enter the correct conference title from your rights confirmation emai (Conference acronym 'XX)*. ACM, New York, NY, USA, 11 pages. https://doi.org/XXXXXXX.XXXXXXX

## 1 INTRODUCTION

Graph neural networks (GNNs) [1, 2] are a class of deep learning models specifically designed to handle graph-structured data, including various scenarios such as social networks [3, 4], finance and insurance technology [5, 6], etc. By harnessing the capabilities of GNNs, social recommendation systems can gain an in-depth understanding of the intricate dynamics and social influence factors that shape users' preferences, leading to improved recommendation accuracy. For example, an insurance company could utilize social relationships extracted from a social network platform by a GNN

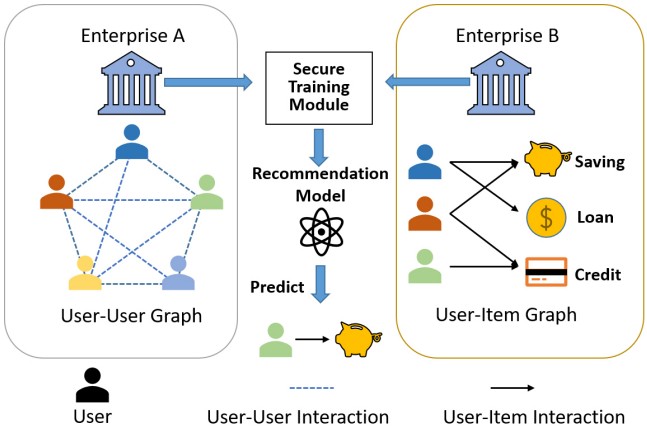

**Figure 1: The example of vertical federated social recommendation with inaccessible social data.**

model to enhance the accuracy of personalized product recommendations (i.e., insurance marketing). However, in real-world scenarios, privacy and business concerns often hinder direct access to private information possessed by aforementioned social platforms. Consequently, the integration of privacy-preserving technologies, such as federated learning [7], secure multi-party computation [8], homomorphic encryption [9], and differential privacy [10], into social recommendation tasks has attracted significant attention from both academia and industries.

Recent works mainly enable the recommender to collaboratively train matrix factorization [11] based recommendation models without accessing the social data owned by other platforms[12, 13]. [12] proposed the secure social MF to utilize the social data as the regularization term when optimizing the model. Further, [13] significantly reduces both the computation and communication costs of the secure social matrix factorization by designing a new secure multi-party computation protocol. However, these solutions cannot be applied to training GNN models, because the computation processes involved in training GNN models are typically more complex compared to MF-based methods. For example, in GNN models, the aggregation of features from different users on the social graph involves multiplying the aggregated results with additional parameter matrices. In contrast, MF-based methods focus on reducing the distances between neighbors' embeddings based on the social data, without the need for additional parameters. In addition, the formulations used in the forward and backward processes of GNN models are much more complex than those of MF-based methods. Consequently, it is essential to develop a secure social recommendation protocol tailored explicitly to enhance the optimization of GNN models.

To address the aforementioned challenges, we propose a novel *vertical federated Social recommendation with Privacy-Preserving Party-to-Party Graph Convolution Networks* (**P4GCN**) to improve the social recommendation system without direct access to the social data. In our approach, we first introduce the *Sandwich-Encryption* module, which ensures data privacy throughout the collaborative computing process. We then provide a theoretical analysis of the security guarantees under the assumption that all participating parties are curious and honest. Finally, extensive experiments are conducted on three real-world datasets, and results demonstrate that our proposed P4GCN outperforms state-of-the-art methods in terms of both recommendation accuracy and communication efficiency.

The main contributions of this study can be summarized as follows:

- We propose P4GCN, a novel method for implementing vertical federated social recommendation with theoretical guarantees. Unlike previous works that assume the availability of social data, we focus on leveraging GNN to enhance recommendation systems with fully unavailable social data in a privacy-preserving manner.
- We introduce the sandwich encryption module, which guarantees data privacy during model training by employing a combination of homomorphic encryption and differential privacy. We provide theoretical guarantees to support its effectiveness.
- Experimental results conducted on four real-world datasets illustrate the enhancements in performance and efficiency. Furthermore, we evaluate the impact of the privacy budget on the utility of the model.

## 2 RELATED WORKS

### 2.1 Social recommendation

Existing social recommendation methods have adopted various architectures according to their goals and achieved outstanding results [14]. For instance, many SocialRS methods employ the graph attention neural network (GANN) [2] to differentiate each user's preference for items or each user's influence on their social friends. Some other methods [15–19] use the graph recurrent neural networks (GRNN) [20, 21] to model the sequential behaviors of users. However, these centralized methods cannot be directly applied when the social data is inaccessible.

### 2.2 Federated recommendation

There are mainly two types of works addressing recommendation systems in FL. The first type is User-level horizontal FL. FedMF [22] safely train a matrix factorization model for horizontal users. FedGNN [23] captures high-order user-item interactions. FedSoG [24] leverages social information to further improve model performance. The second type is Enterprise-level vertical FL which considers training a model with separated records kept by different companies. To promise data security in this case, techniques such as differential privacy[25] and homomorphic encryption[26], are widely used. [27] uses random projection and ternary quantization mechanisms to achieve outstanding results in privacy-preserving.

However, these works failed to construct the social recommendation model when the social data is unavailable. To address this issue, SeSorec[12] protects social information while utilizing the social data to regularize the model. [13] proposed two secure computation protocols to further improve the training efficiency. Although these works can be applied to matrix factorization models, the GNN-based models have not been considered in this case.

## 3 PROBLEM FORMULATION

In this section, we first introduce the notations we used, and then we give the formal definition of our problem. Let $U = \{u_i\}, u_i \in \mathbb{N}$ denote the user set and $V = \{v_i\}, v_i \in \mathbb{N}$ denote the item set, where the number of users is $N_U = |U|$ users and the number of items is $N_V = |V|$. There are two companies $\mathcal{P}_1, \mathcal{P}_2$ that own different parts of the user and item data. $\mathcal{P}_1$ owns the user set $U$ and the item set $V$ with the interactions between users and items $\mathcal{R} = \{(u_i, v_j, r_k)\}$, where each $r_k \in \mathbb{R}$ is a scalar that describes the $k$th interaction in $\mathcal{R}$. $\mathcal{P}_2$ owns the same user set $U$ and their social data (i.e. user-user interactions) $\mathcal{S} = \{(u_i, u_j, s_k)\}$, where $s_k \in \mathbb{R}$ denotes the $k$th interaction in $\mathcal{S}$.

$\mathcal{P}_1$ and $\mathcal{P}_2$ collaboratively train a social recommendation GNN-based model $f_{\boldsymbol{\theta}}$ that predicts the rating $\hat{r}_{u_i v_j} = f(U, V, \mathcal{R}_{train}, \mathcal{S})$ of the user $u_i$ assigning to the item $v_j$. We minimize the mean square errors (i.e. MSE) [12] between the predictions and the targets to optimize the model parameters $\boldsymbol{\theta}$:

$$\min_{\boldsymbol{\theta}} \mathcal{L}(\boldsymbol{\theta}; U, V, \mathcal{R}_{train}, \mathcal{S}) = \frac{1}{|\mathcal{R}_{train}|} \sum_{(u_i, v_j, r_k) \in \mathcal{R}_{train}} \|r_k - \hat{r}_{u_i v_j}\|^2$$

Since all the computation can be done by $\mathcal{P}_1$ itself except for the GNN layers for the social aggregation, we focus on protecting data privacy when computing the results of the social aggregation layer. Particularly, we consider the most classical GNN operator, Graph Convolution (GC), as the social aggregation operator in our model. Given a social-aggregation GC operator $GC(\mathbf{X}, \mathbf{A}, \boldsymbol{\theta}_{GC})$, $\mathcal{P}_1$ should realize message passing mechanism of user features $\mathbf{X} \in \mathbb{R}^{N \times d}$ over the users' social graph $\mathbf{A} \in \{a_{ij}\}_{N \times N}, a_{ij} \in \{0, 1\}$ (i.e. the adjacent matrix) as below:

**Forward.**
$$\tilde{\mathbf{L}}_{sym} = \mathbf{D}^{-\frac{1}{2}}(\mathbf{A} + \mathbf{I})\mathbf{D}^{-\frac{1}{2}}, \mathbf{D} = \text{diag}([1 + \sum_j a_{1j}, ..., 1 + \sum_j a_{Nj}])$$
(1)

$$\mathbf{Z} = \sigma(\mathbf{Y} + \mathbf{1}\mathbf{b}^{\top}), \mathbf{Y} = \tilde{\mathbf{L}}_{sym}\mathbf{X}\mathbf{W} \tag{2}$$

**Backward.**
$$\frac{\partial \mathcal{L}}{\partial \mathbf{X}} = \frac{\partial \mathcal{L}}{\partial \mathbf{Y}}\frac{\partial \mathbf{Y}}{\partial \mathbf{X}} = \tilde{\mathbf{L}}_{sym}\frac{\partial \mathcal{L}}{\partial \mathbf{Y}}\mathbf{W}^{\top}, \frac{\partial \mathcal{L}}{\partial \mathbf{W}} = \frac{\partial \mathcal{L}}{\partial \mathbf{Y}}\frac{\partial \mathbf{Y}}{\partial \mathbf{W}} = \mathbf{X}^{\top}\tilde{\mathbf{L}}_{sym}^{\top}\frac{\partial \mathcal{L}}{\partial \mathbf{Y}}$$
(3)

where the parameters of graph convolution are $\boldsymbol{\theta}_{GC} = [\mathbf{W}; \mathbf{b}], \mathbf{W} \in \mathbb{R}^{d_{in} \times d_{out}}, \mathbf{b} \in \mathbb{R}^{d_{out}}$. We consider safely computing the two processes under the limitation that data privacy should be bi-directionally protected for these processes, where the parties cannot have access to another one's data (i.e. $\mathcal{P}_1$ cannot infer the adjacent matrix $\mathbf{A}$ and $\mathcal{P}_2$ cannot infer the node features $\mathbf{X}$ during computation). We follow [12] to assume that all the parties are honest and curious. Different from works that consider each party to own user-user and user-item interactions partially, we attempt to apply GNN

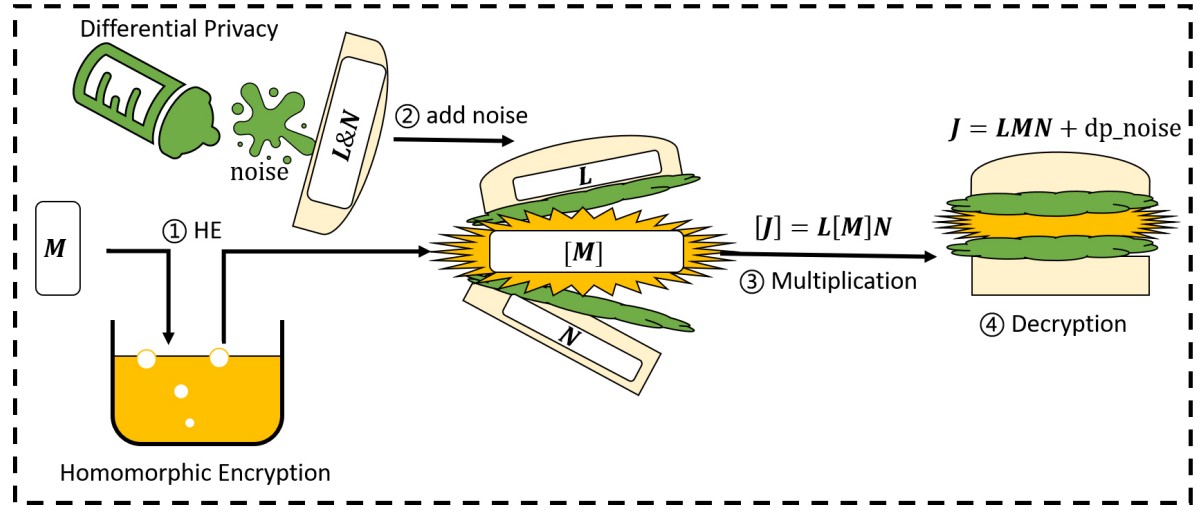

**Figure 2: The framework of the *Sandwich Encryption***

.

modules to the **social-data-fully-inaccessible** vertical federated social recommendation.

## 4 METHODOLOGY

### 4.1 Motivation

After social aggregation in Eq.(1) and Eq.(2), $\mathcal{P}_1$ obtains the output $Y$ for further computation of the loss $\mathcal{L}$. To optimize the model, $\mathcal{P}_1$ uses $\frac{\partial \mathcal{L}}{\partial Y}$ to compute the derivate of node features $\frac{\partial \mathcal{L}}{\partial X}$. We notice that a key computation paradigm, multiplying three matrices, repeatedly appears in both forward and backward processes. Further, if we let the parameter matrix $W$ be kept by $\mathcal{P}_2$ that owns $\tilde{L}_{sym}$, the matrices on both sides and the matrix at the middle for each equation will be kept by different parties. In addition, the left-side result of each equation will be only needed by the one that owns the middle matrix. This observation motivates us to consider such a problem

*Given the matrices* $L \in \mathbb{R}^{p \times q}, N \in \mathbb{R}^{r \times s}$ *owned by the party* $p_1$ *and the matrix* $M \in \mathbb{R}^{q \times r}$ *owned by the party* $p_2$*, how can we design an algorithm to satisfy the two requirements below*

**R1.** *the party* $p_2$ *obtains the multiplication* $J = LMN$ *without exposing* $M$ *to the party* $p_1$.
**R2.** *the party* $p_2$ *cannot infer* $L$ *and* $N$ *from* $J$ *and* $M$.

As long as the above problem is solved, the computing processes of a graph convolution operator can be done without leaking data privacy. Therefore, we now focus on how to find a solution to this problem with the theoretical guarantee of privacy-preserving.

### 4.2 Sandwich encryption

*4.2.1 Solution to R1.* For the first requirement, each time there is a need to compute $J = LMN$, the party first $p_2$ encrypts the matrix $M$ with the public key $\mathcal{P}_{pub,2}$ by simply using *Homomorphic Encryption* (e.g. Paillier [28]). Then, the ciphertext $[M]_{\mathcal{P}_{pub,2}}$ is sent to the party $p_1$ to compute $[J]_{\mathcal{P}_{pub,2}} = L[M]_{\mathcal{P}_{pub,2}}N$, and the result

is returned to $p_2$. By decrypting the result with the private key $\mathcal{P}_{prv,2}$, $p_2$ can know $J$ without leaking $M$ to $p_1$.

*4.2.2 Solution to R2.* Now we discuss how to protect privacy for $L$ and $M$.

*Database-level protection.* Since $p_2$ doesn't know the exact values of both the two side matrices, it brings significant challenges for $p_2$ to steal information about them from $J$ and $M$. To better illustrate this, we take an example where all variables of the equation $j = lmn$ are scalars, and we can thus infer that $j/m = ln$, which indicates there are infinite combinations of $l$ and $n$ for any given $j \neq 0, m \neq 0$. For the matrix case, we illustrate the protection on the database level through Theorem 1.

**THEOREM 4.1.** *Given* $J = LMN$ *where all matrices are not zero matrices, there exists infinite combinations of* $N' \neq N, L' \neq L$ *such that* $J = L'MN'$.

PROOF. See Appendix A.2. □

Therefore, without knowing $L$ (or $N$), $p_2$ cannot fully recover $N$ (or $L$), leading to the *database-level* privacy protection. However, this barrier fails to protect the privacy of the two-side matrices at the element level. For example, if there are only two users' embeddings in $M \in \mathbb{R}^{2 \times d_{in}}$ and one of the two embeddings happens to be zero, we can easily infer whether the two users have social interactions from the result $J \in \mathbb{R}^{2 \times d_{out}}$ by recognizing whether the aggregated embeddings corresponding to the zero embedding are still zero.

*Element-level protection.* To further enhance privacy protection for the two-side matrices at the element level, we introduce differential privacy (DP) noise [10] to the computed result $J$. DP offers participants in a database the compelling assurance that information from datasets is virtually indistinguishable whether or not someone's personal data is included. Since the object to be protected can be of high dimension, we leverage the advanced matrix-level DP

**Figure 3: The training workflow of the proposed P4GCN**

.

---

**Algorithm 1** *Sandwich Encryption Framework*

---

1: **Input:** The party $p_1$ owning $(\mathbf{L}, \mathbf{N})$, the party $p_2$ owning $\mathbf{M}$, differential privacy process $g_{dp}(\cdot)$, the key pair $< \mathscr{P}_{pub,2}, \mathscr{P}_{prv,2} >$ of $p_2$

2: **Out:** $\mathbf{J}'$ to $p_2$

3: $p_2$ encrypts $\mathbf{M}$ with its public key $\mathscr{P}_{pub,2}$ to obtain $[\mathbf{M}]$ by *Homomorphic Encryption*, and send it to $p_1$.

4: $p_1$ calculate $[\mathbf{J}'] = g_{dp}(\mathbf{L}, \mathbf{N}, [\mathbf{M}])$ such that $[\mathbf{J}'] = \mathbf{L}[\mathbf{M}]\mathbf{N} + \epsilon_{dp}$, and send $[\mathbf{J}']$ to $p_2$.

5: $p_2$ decrypts $[\mathbf{J}']$ with its private key $\mathscr{P}_{prv,2}$ to obtain $\mathbf{J}'$.

---

mechanism, aMGM, introduced by [29, 30] to enhance the utility of the computation.

*Definition 4.2 (analytic Matrix Gaussian Mechanism [29]).* For a function $f(\mathbf{X}) \in \mathbb{R}^{m \times n}$ and a matrix variate $\mathbf{Z} \sim \mathcal{MN}_{m,n}(\mathbf{0}, \Sigma_1, \Sigma_2)$, the analytic Matrix Gaussian Mechanism is defined as

$$\text{aMGM}(f(\mathbf{X})) = f(\mathbf{X}) + \mathbf{Z} \qquad (4)$$

where $\mathcal{MN}_{m,n}(\mathbf{0}, \Sigma_1, \Sigma_2)$ denotes matrix gaussian distribution .

*Definition 4.3 (Matrix Gaussian Distribution[30]).* The probability density function for the $m \times n$ matrix-valued random variable $\mathbf{Z}$ which follows the matrix Gaussian distribution $\mathcal{MN}_{m,n}(\mathbf{M}, \Sigma_1, \Sigma_2)$ is

$$\Pr(\mathbf{Z}|\mathbf{M}, \Sigma_1, \Sigma_2) = \frac{\exp \frac{1}{2}\|\mathbf{U}^{-1}(\mathbf{Z} - \mathbf{M})\mathbf{V}^{-\top}\|_F^2}{(2\pi)^{mn/2}|\Sigma_2|^{n/2}|\Sigma_1|^{m/2}} \qquad (5)$$

where $\mathbf{U} \in \mathbb{R}^{m \times m}, \mathbf{V} \in \mathbb{R}^{n \times n}$ are invertible matrices and $\mathbf{U}\mathbf{U}^\top = \Sigma_1, \mathbf{V}\mathbf{V}^\top = \Sigma_2$. $|\cdot|$ is the matrix determinant and $\mathbf{M} \in \mathbb{R}^{m \times n}, \Sigma_1 \in \mathbb{R}^{m \times m}, \Sigma_2 \in \mathbb{R}^{n \times n}$ are respectively the mean, row-covariance, column-covariance matrices.

The privacy protection is guaranteed by Lemma.4.4

LEMMA 4.4 (DP OF AMGM [29]). *For a query function $f$, aMGM satisfies $(\epsilon, \delta) - DP$, iff*

$$\frac{s_2(f)}{b} \leq \sigma_m(\mathbf{U})\sigma_n(\mathbf{V}) \qquad (6)$$

*where $b$ is decided by $(\epsilon, \delta)$ and $s_2(f)$ is the $L_2$-sensitivity, $\sigma_m(\mathbf{U})$ and $\sigma_n(\mathbf{V})$ are respectively the smallest singular values of $\mathbf{U}$ and $\mathbf{V}$.*

The general procedure of the *Sandwich Encryption* is listed in Algorithm.1. The encryption process is like making a sandwich where the two pieces of bread are corresponding to the two-side matrices and the middle matrix is the meat in the sandwich as shown in Figure 2. By properly pre-processing the materials, the data privacy of each material can be preserved. While we apply DP to enhance privacy protection, how to preserve the utility of these computing processes as much as possible still brings non-trivial challenges. To this end, we design the Privacy-Preserving Two-Party Graph Convolution Network (P4GCN) to enhance the utility of the model while applying DP.

## 4.3 P4GCN

### 4.3.1 Architecture.

*Overview.* The architecture of P4GCN is as shown in Figure 3. During each training iteration, $\mathcal{P}_1$ first locally aggregates the user features $\mathbf{X}_{user}^{(0)}$ and the item features $\mathbf{X}_{item}^{(0)}$ by the backend (e.g., LightGCN[31]) into embeddings $\mathbf{X}_{user}^{(1)}$ and $\mathbf{X}_{item}^{(1)}$. Then, $\mathcal{P}_1$ uses Algo.1 to collaboratively compute the user social embeddings that are aggregated on the social data by the GCN layer with $\mathcal{P}_2$. After obtaining the user social embeddings $\mathbf{X}_{user}^{(2)}$, $\mathcal{P}_1$ uses the fusion layer to aggregate $\mathbf{X}_{user}^{(1)}$ and $\mathbf{X}_{user}^{(2)}$ together to construct the new

user embeddings $\mathbf{X}_{user}^{(3)}$. Finally, both $\mathbf{X}_{user}^{(3)}$ and $\mathbf{X}_{item}^{(1)}$ are input into the decoder to obtain the predictions to compute the loss. The backward computation of the social GCN layer is also protected by Algo.1.

*Fusion Layer.* The fusion layer is designed for two reasons. For one thing, the DP mechanism may bring too much noise that leads to the degradation of the model performance. For another thing, the social information of different users may not consistently improve the model's performance but harm it. Therefore, we design the fusion layer to adaptively extract useful information by reweighing the inputs. Concretely, the fusion layer allocates weights to each activation in each user's embeddings by a two-layer MLP with a softmax function and position-wisely fuses them. This introduces a chance for the party $\mathcal{P}_1$ to avoid the collaboration significantly reducing local model performance.

*4.3.2 Privacy-Preserved Social Aggregation.* We analyze the sensitivity of the graph convolution and then apply aMGM to its computing processes.

*Forward.* During aggregation, the user $i$'s social embedding is specified by $\mathbf{x}_i^{(2)} = \mathbf{X}_{user,\cdot i}^{(2)} = \mathbf{l}_i \mathbf{X} \mathbf{W}, \mathbf{l}_i = \tilde{\mathbf{L}}_{sym,\cdot i}$, which can be independently computed without queries on other users' social embeddings. Therefore, we focus on the computing sub-process $f_i(\mathbf{l}_i, \mathbf{X}, \mathbf{W})$ to protect user-level privacy (i.e., the social interaction between any two users). Given two adjacent social databases $\mathbf{A}$ and $\mathbf{A}'$ whose elements are the same except one (e.g., $\|\mathbf{A} - \mathbf{A}'\|_F = 1$), the $L_2$-sensitivity of each $f_i, \forall i \in [N]$ is bounded by

$$s_2(f_i) = \max_{A,A'} \|\mathbf{l}_i' \mathbf{X} \mathbf{W} - \mathbf{l}_i \mathbf{X} \mathbf{W}\|_F \leq \|\mathbf{X}\|_F \|\mathbf{W}\|_F s_l(i) \quad (7)$$

$$s_l(i) = \begin{cases} (\frac{1}{2} + \frac{1}{2}c_o)^{1/2}, & , \mathbf{a}_i = \mathbf{0} \\ (\frac{1}{\|\mathbf{a}_i\|_1^2 + \|\mathbf{a}_i\|_1} c_i + \frac{1}{\|\mathbf{a}_i\|_1} c_o)^{1/2}, & , \text{else} \end{cases} \quad (8)$$

where $c_i = \sum_{j=1}^{N} \frac{a_{ij} + \mathbf{1}(i=j)}{\|\mathbf{a}_j\|_1 + 1} \leq \|\mathbf{a}_i\|_1 + 1, c_o = \max_j \frac{1}{\|\mathbf{a}_j\|_1 + 1} \leq 1, s_l(i) \leq 2$ always hold for all users. Then, we respectively clip $\mathbf{X}$ and $\mathbf{W}$ by $\max(1, \frac{\|\cdot\|_2}{C})$ to bound the sensitivity $s_2(f_i) \leq C^2 s_l(i)$ (e.g., the coefficient $C = 1$ in experiments) before computation and finally rescale the computed result by the inverse scale factor. We empirically scale $\tilde{\mathbf{L}}_{sym}$ with a factor $\frac{1}{N}$ in practice. We detail the derivation of the sensitivity term in Appendix A.1.

*Backward for node features.* The backward process for node features $f_i^{back}$ is $\eta \frac{\partial \mathcal{L}}{\partial \mathbf{x}_i^{(2)}} = \mathbf{l}_i (\eta \frac{\partial \mathcal{L}}{\partial \mathbf{Y}}) \mathbf{W}^\top$. We bound the sensitivity of $f_i^{back}$ like forward process $f_i$, leading to the same bound

$$s_2(f_i^{back}) \leq C^2 s_l(i) \quad (9)$$

*Backward for model parameters.* The backward process for model parameters is $\frac{\partial \mathcal{L}}{\partial \mathbf{W}} = \mathbf{X}^\top \tilde{\mathbf{L}}_{sym}^\top \frac{\partial \mathcal{L}}{\partial \mathbf{Y}}$. We notice that the actual function sensitivity can be significantly influenced by the Frobenius norms of all the three matrices that scale with the user number $N$, leading to large noise added to the computed result. Therefore, we seek for an alternative to this computing process by splitting $\mathbf{W} = \mathbf{W}_{\mathcal{P}_2} \mathbf{W}_{\mathcal{P}_1}, \mathbf{W}_{\mathcal{P}_2} \in \mathbb{R}^{d_{in} \times d_{in}}, \mathbf{W}_{\mathcal{P}_1} \in \mathbb{R}^{d_{in} \times d_{out}}$ and freeze $\mathbf{W}_{\mathcal{P}_2}$ that is kept by the party $\mathcal{P}_2$ without updating it. We initialize $\mathbf{W}_{\mathcal{P}_2}$ by normal distribution to approximate full rank and then

**Table 1: Dataset statistics**

| Dataset | CiaoDVD | FilmTrust | Douban | Epinions |
|---|---|---|---|---|
| **Users** | 7375 | 1508 | 3000 | 22158 |
| **Items** | 99746 | 2071 | 3000 | 296277 |
| **Ratings** | 278483 | 35497 | 136891 | 728517 |
| **Social Links** | 111781 | 1853 | 7765 | 355364 |
| **Density**$_{Rating}$ | 0.0379% | 1.1366% | 1.5210% | 0.0110% |
| **Density**$_{Link}$ | 0.2055% | 0.0815% | 0.0863% | 0.0723% |

clip it only once before training starts. The parameter $\mathbf{W}_{\mathcal{P}_1}$ is updated by $\mathcal{P}_1$ without any communication to $\mathcal{P}_2$ since components in $\frac{\partial \mathcal{L}}{\partial \mathbf{W}_{\mathcal{P}_1}} = (\tilde{\mathbf{L}}_{sym} \mathbf{X} \mathbf{W}_{\mathcal{P}_1})^\top \frac{\partial \mathcal{L}}{\partial \mathbf{Y}}$ are already known by $\mathcal{P}_1$.

*Privacy.* We independently apply aMGM mechanism to each user's social embedding based on its sensitivity bound (e.g., Eq.(4.3.2) and Eq.(9)). Eq.(8) suggests that the more social relations one user owns, the smaller sensitivity its computing process is, resulting in less noise being injected into the intermediates of this user. The total privacy cost can be estimated by the maximum privacy cost among users according to the parallel composition theorem [32]. We follow [29] to accumulate privacy costs across iterations based on the privacy loss distribution of aMGM in Lemma.4.5.

LEMMA 4.5. *[Privacy Loss of aMGM.[30]] The privacy loss variable of aMGM follows gaussian distribution $\mathcal{N}(\eta, 2\eta)$ and $\eta$ is given by $\eta = \frac{\|\mathbf{U}^{-1}(f(\mathbf{X}) - f(\mathbf{X}'))\mathbf{V}^{-\top}\|_F^2}{2}$.*

*4.3.3 Efficiency.*

*Batch-wise optimization.* We now show how to optimize the model in a batch-wise manner for efficiency. The full batch training will bring large communication and computation costs (e.g., frequently encrypting large matrices and transmitting the expanded ciphertext). To tackle this issue, for a batch of records $\{(u_{b_k}, v_{b_k}, r_{b_k})\}|_{k=1}^{|B|}$, we denote the users in the current batch as $\mathcal{B} \in \mathbb{R}^{|\mathcal{B}| \times N}, |\mathcal{B}| \leq |B|$. Then, the corresponding computing process is

$$\mathbf{Y}_B = (\mathcal{B}\tilde{\mathbf{L}}_{sym}) \mathbf{X} \mathbf{W}, \frac{\partial \mathcal{L}}{\partial \mathbf{X}_B} = (\mathcal{B}\tilde{\mathbf{L}}_{sym} \mathcal{B}^\top) \frac{\partial \mathcal{L}}{\partial \mathbf{Y}_B} \mathbf{W}^\top \quad (10)$$

In this way, the party $\mathcal{P}_2$ can store the full ciphertext $[\mathbf{X}]$ that will be only encrypted once and batch-wisely update it by $\eta[\frac{\partial \mathcal{L}}{\partial \mathbf{X}_B}]$. Unlike full batch training, the embeddings of users out of the batch cannot be updated. Otherwise, the social interactions will be easily exposed to the recommender.

*Communication.* The communication cost lies in the transmission of the encrypted middle matrices (i.e. $\mathbf{X}, \frac{\partial \mathcal{L}}{\partial \mathbf{Y}_B}$) and the results (i.e. $\mathbf{Y}_B, \frac{\partial \mathcal{L}}{\partial \mathbf{X}_B}$). Since $\mathbf{X}$ is only encrypted and transmitted once, the total communication cost is $O(Nd + TBd)$ over iterations $T$ where $\frac{\partial \mathcal{L}}{\partial \mathbf{Y}_B}, \mathbf{Y}_B \in \mathbb{R}^{|\mathcal{B}| \times d_{out}}, \frac{\partial \mathcal{L}}{\partial \mathbf{X}_B} \in \mathbb{R}^{|\mathcal{B}| \times d_{in}}$ and $d = \max(d_{in}, d_{out})$.

# 5 EVALUATION

## 5.1 Experimental Setting

**Table 2: Comparison results of different models in terms of model accuracy (in RMSE and MAE). The optimal (second optimal) result of each column is bolded (underlined).**

| Method | | FilmTrust | | CiaoDVD | | Douban | | Epinions | |
|---|---|---|---|---|---|---|---|---|---|
| | | RMSE | MAE | RMSE | MAE | RMSE | MAE | RMSE | MAE |
| Local | PMF | 0.8007 | 0.6106 | 1.2245 | 0.9651 | 0.8361 | 0.6300 | 1.2487 | 0.9721 |
| | NeuMF | 0.8287 | 0.6319 | 1.1842 | 0.8839 | 0.7894 | 0.6222 | 1.1285 | **0.8020** |
| | GCN | 0.8765 | 0.6796 | 1.1076 | 0.8383 | 0.7989 | 0.6346 | 1.1513 | 0.8177 |
| | LightGCN | 0.7960 | 0.6079 | 1.1186 | 0.8396 | 0.7892 | 0.6209 | 1.0746 | 0.8412 |
| | FeSog⁻ | 0.8029 | 0.6118 | 1.2314 | 0.9741 | 0.8331 | 0.6498 | 1.2171 | 0.9530 |
| Social | SeSoRec | 0.8009 | 0.6106 | 1.1988 | 0.9635 | 0.8171 | 0.6316 | 1.2131 | 0.9598 |
| | S3Rec | 0.8009 | 0.6106 | 1.1988 | 0.9635 | 0.8171 | 0.6316 | 1.2131 | 0.9598 |
| | P4GCN | 0.7929 | 0.6059 | **1.0776** | **0.8224** | 0.7672 | **0.6023** | 1.0744 | 0.8272 |
| | P4GCN* | **0.7905** | **0.6032** | 1.0803 | 0.8225 | **0.7670** | 0.6035 | **1.0642** | 0.8186 |

*Datasets.* We use four social recommendation datasets to validate the effectiveness of the proposed method: Filmtrust [33], CiaoDVD [34], Douban [35], and Epinions [36]. Specifically, we set the social data owned by $\mathcal{P}_2$ and other data owned by $\mathcal{P}_1$. We show the statistics of the datasets in Table 1.

*Implementation.* All our experiments are implemented on a Ubuntu 16.04.6 server with 64 GB memory, 4 Intel(R) Xeon(R) CPU E5-2630 v4 @ 2.20GHz, 4 NVidia(R) 3090 GPUs, and PyTorch 1.10.1.

*Baselines.* We compare P4GCN with two types of baselines. The first type contains traditional methods without using social data. These methods are concluded as follows

- **PMF**[11] is a classic matrix factorization model that only uses rating data on $\mathcal{P}_1$.
- **NeuMF**[37] is a neuron-network-based matrix factorization method that has superior performance against traditional MF methods.
- **GCN**[38] is a classic convolutional graph neural network that only uses rating data on $\mathcal{P}_1$.
- **LightGCN**[31] improves the convolutional graph neural network by reducing the parameters and aggregating the activations of different layers.
- **FeSog⁻**[24] removes the social aggregation module from the original version that requires social links to be stored together with user features, which will break our fundamental assumption of inaccessible social data. We compare FeSog with fully available data in Sec. 5.7

The second type contains methods that safely use social data to make social recommendations:

- **SeSoRec**[12] tries to solve the privacy-preserving cross-platform social recommendation problem, but suffers from security and efficiency problems.
- **S³Rec**[13] is the state-of-the-art method that solves the safety problem and improves the efficiency within the scope of matrix factorization on the basis of **SeSoRec**.

- **P4GCN** (ours) is set to satisfy $(\epsilon, \delta)$-DP guarantee (e.g., $\epsilon$ depends on the dataset) and **P4GCN*** corresponds to the ideal case without injecting DP noise.

*Hyper-parameters.* We fix the embedding dimensions $k = 64$ of the model for all the datasets. We tune the learning rate $\eta \in \{1e-3, 1e-2, 1e-1, 1, 10, 100, 1000\}$ and batch size $|B| \in \{64, 256, 512, 1024, 2048, 4096, \text{full}\}$ to achieve each method's optimal results. We respectively limit the privacy budgets of P4GCN by $\epsilon = \{15.0, 10.0, 10.0, 3.0\}$ and $\delta = 1e-4$ across datasets in columns of Table 2 (i.e., FilmTrust, CiaoDvd, Douban, and Filmtrust). The hyper-parameter $\beta_{\text{P4GCN}}$ is tuned on $\{0.01, 0.05, 0.1, 0.5, 1.0, 10.0, 100.0\}$ and both $\lambda_{\text{SeSoRec}}$ and $\lambda_{\text{S3Rec}}$ are tuned on $\{1e-4, 1e-3, 1e-2, 1e-1\}$.

*Metrics.* We follow previous works [4] to use Root Mean Square Error (RMSE) and Mean Absolute Error (MAE) as the evaluation metrics of model performance.

## 5.2 Model performance

From Table 2, we find that: (1) P4GCN* without DP consistently improves both MAE and RMSE metrics over all the baselines on the first three datasets (i.e., FilmTrust, CiaoDVD, and Douban) and achieves competitive results (e.g., RMSE= 1.0642, MAE=0.8186) against others' optimal results (e.g., $\text{RMSE}_{\text{LightGCN}} = 1.0746$ and $\text{MAE}_{\text{NeuMF}} = 0.8020$). (2) Our proposed Sandwich Encryption Module can well preserve the final model performance over four datasets given proper privacy budges, which achieves the optimal or second optimal results over 87.5% columns. (3) P4GCN exhibits superior performance to traditional matrix-decomposition-based social recommendation (e.g., SeSoRec and S3Rec), especially on datasets of large-scale (e.g., CiaoDVD with 7375 clients and Epinions with 22158 clients). We attribute this enhancement to the adaption of GNN which has a stronger representation ability than the traditional matrix-decomposition-based model in recommendation.

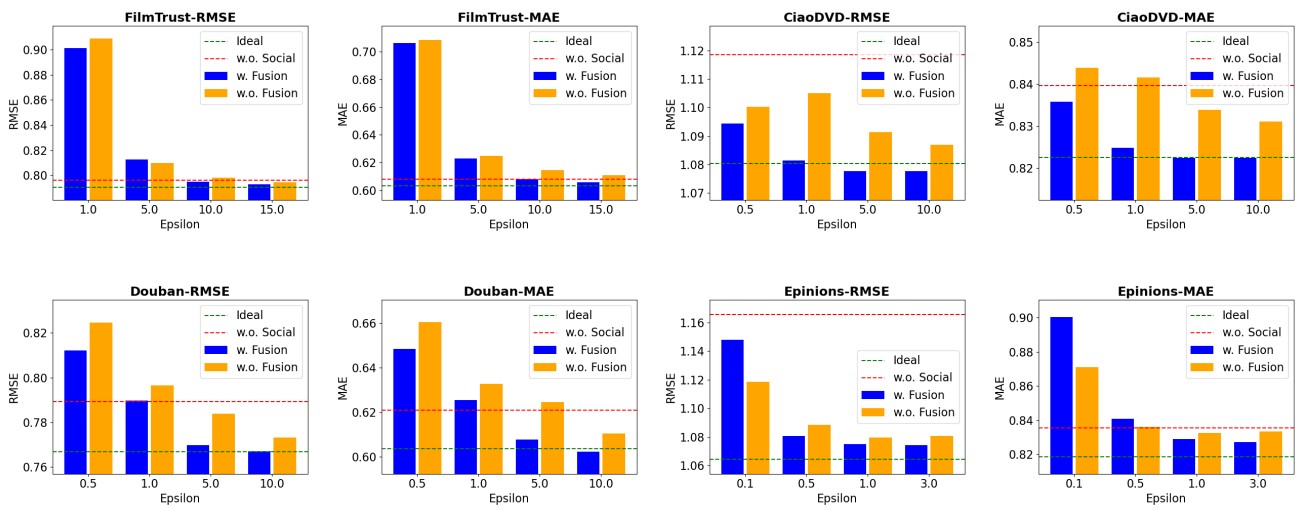

Figure 4: The model performance RMSE and MAE of P4GCN w/w.o. fusion layer v.s. privacy budget $\epsilon$.

## 5.3 Impact of privacy budget $\epsilon$

*Privacy Budget.* We investigate the impact of privacy budget $\epsilon$ on our proposed method in Figure 4, where the red dashed line corresponds to results without leveraging social data and the green dashed line corresponds to the ideal results without adding DP noise. First, as the privacy budget grows properly, P4GCN introduces non-trivial improvements over the results without using social information (e.g., the bars below the red dashed lines). Second, our proposed privacy-preserving mechanism can well preserve the performance of the ideal case without adding DP noise (e.g., the green dashed lines), which confirms the effectiveness of our P4GCN in leveraging social data to enhance existing recommendation systems.

*Ablation on the fusion layer.* We further demonstrate the effectiveness of the fusion layer integrated into P4GCN by directly averaging the user social embeddings (e.g., scaled by $\beta$) and the original user embeddings for comparison. As shown in Figure 4, P4GCN will suffer performance degradation after removing the fusion layer across different datasets, where most of the yellow bars are higher than the blue ones under the same privacy budget $\epsilon$. In addition, P4GCN w.o. the fusion layer failed to approximate the ideal performance even though the privacy budget is relatively large (e.g., $\epsilon = 10.0$ in CiaoDVD), while the version w. Fusion did. This suggests the excellent ability of the fusion layer to aggregate the social information into the user features. Further, P4GCN with the fusion layer also shows a better tolerance to the low privacy budget than the one without using the fusion layer. For example, P4GCN w.o. the fusion layer will harm the original recommendation system on FilmTrust when $\epsilon = 10.0$ and Douban when $\epsilon = 5.0$, while the usage of the fusion layer decreases the minimal effective privacy budget. These results confirm the effectiveness of the proposed fusion layer in both handling DP-noise and fusing social information.

Table 3: The improvement on model performance by integrating P4Layer to existing methods.

| Method | | FilmTrust | | CiaoDVD | |
|---|---|---|---|---|---|
| | | RMSE | MAE | RMSE | MAE |
| **PMF** | **original** | 0.8007 | 0.6106 | 1.2245 | 0.9651 |
| | **+P4Layer&DP** | **0.7997** | 0.6112 | 1.2163 | 0.9648 |
| | **+P4Layer-Ideal** | **0.7997** | **0.6105** | **1.2125** | **0.9642** |
| **GCN** | **original** | 0.8765 | 0.6796 | 1.1709 | 0.8731 |
| | **+P4Layer&DP** | 0.8569 | 0.6606 | **1.1388** | 0.8766 |
| | **+P4Layer-Ideal** | **0.8506** | **0.6486** | 1.1414 | **0.8598** |

## 5.4 Integrate To Existing Methods

We show that existing local recommendation methods (e.g., PMF and GCN) without considering social data can benefit from our proposed P4Layer on FilmTrust and CiaoDVD in Table 3, which suggests that companies can improve their local recommendation system by leveraging our proposed P4GCN in a plug-in manner. The parameters of differential privacy are consistent with the settings in Table 2.

## 5.5 Impact of hyper-parameter $\beta$

We study the impact of the choice of hyper-parameter $\beta$ on the model performance in Figure 5. We denote P4GCN without adding DP noise as the ideal case (e.g., the red notations). The figure shows that the optimal value of $\beta$ is always larger than 0 across all the datasets, indicating that the recommendation system can consistently benefit from social information integrated by our P4GCN regardless of differential privacy. In addition, the DP noise lowers the optimal degree of leveraging social information (e.g., the blue

**Table 4: Communication costs (GB) under the fixed epoch $E = 5$ with varying batch sizes (e.g., 64, 1024, and 4096) and the practical cost in our experiments in Table 2 (e.g., the last column)**

| Dataset | Method | B=64 | B=1024 | B=4096 | Prac. |
|---------|--------|------|--------|--------|-------|
| FilmTrust | P4GCN | 10.70 | 10.68 | 3.81 | 61.77 |
| | S3Rec | 5.48 | 5.47 | 1.78 | 118.33 |
| CiaoDVD | P4GCN | 21.88 | 21.88 | 21.74 | 21.88 |
| | S3Rec | 15.01 | 15.01 | 14.88 | 21.00 |
| Douban | P4GCN | 42.28 | 42.22 | 31.44 | 82.18 |
| | S3Rec | 19.23 | 19.20 | 14.01 | 33.70 |
| Epinions | P4GCN | 394.44 | 394.44 | 394.24 | 716.38 |
| | S3Rec | 1160.98 | 1160.96 | 1160.09 | 2785.83 |

star never appears on the left of the red star) since the aggregation efficiency can be degraded by the noise. We also notice that a large value of $\beta$ will lead to the degradation of the model performance, which suggests the choice of $\beta$ should be very careful in practice. We consider how to efficiently and adaptively decide effective $\beta$ as our future works.

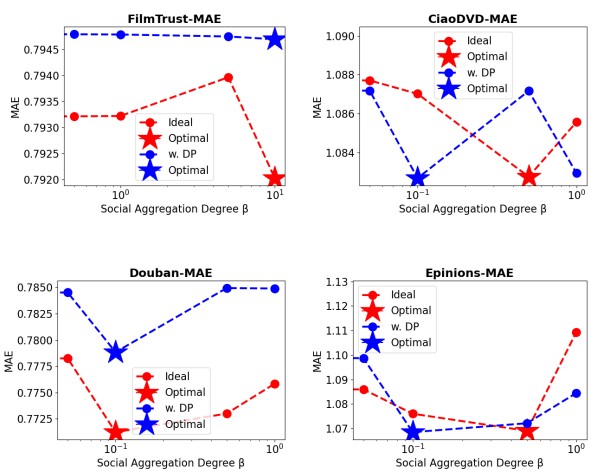

**Figure 5: The impact of social aggregation degree $\beta$ v.s. model performance (i.e. MAE)**

### 5.6 Communication cost

We list the communication costs of P4GCN and another communication-efficient VFL social recommendation method (i.e., S3Rec [13]) in Table 4. We report the communication costs under fixed parameter settings (e.g., 3th-5th columns) and the practical settings used in Table 2 (e.g., the last columns). P4GCN causes nearly 2.2× costs than S3Rec when the epoch number and batch size are fixed on three datasets (i.e., FilmTrust, CiaoDVD, and Douban), and P4GCN saves

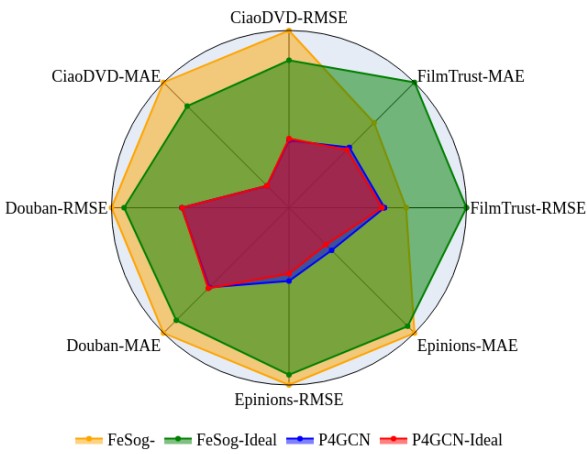

**Figure 6: The model performance of FeSog and P4GCN across datasets where smaller areas are better. Each metric is divided by its corresponding maximum value for a clear view.**

$\frac{2}{3}$ costs on Epinions. Although S3Rec exhibits lower communication amounts than P4GCN under fixed settings, P4GCN can achieve competitive communication efficiency when each method runs until reaching its optimal results. We also plan to further improve the communication efficiency of P4GCN in our future works.

### 5.7 Comparison with FeSog w. social data

We finally compare our method with FeSog-Ideal which can directly access the full social data to verify the advantage of P4GCN in enhancing recommendation systems with social data. As shown in Figure 6, integrating social data can slightly improve model performance in FeSog when the social data is fully available in most cases (e.g., CiaoDVD, Douban, and Epinions). However, FeSog-Ideal failed to leverage social data to enhance performance in FilmTrust. We attribute this to the weak connection between social information and recommendations in FilmTrust, where S3Rec/SeSoRec also suffers similar failure and the improvement of P4GCN is also limited. Further, our P4GCN dominates FeSog in terms of RMSE and MAE across all the datasets regardless of the availability of social data to FeSog and the usage of differential privacy, which confirms the advantage of P4GCN in federated social recommendation.

### 6 CONCLUSION

This paper addresses the development of GNN-based models for a secure social recommendation. We present P4GCN, a novel vertical federated social recommendation approach designed to enhance recommendation accuracy when dealing with inaccessible social data. P4GCN incorporates a sandwich-encryption module, which guarantees comprehensive data privacy during collaborative computing. Experimental results on four datasets demonstrate that P4GCN outperforms state-of-the-art methods in terms of recommendation accuracy. We are considering leveraging other formats of graph information like LLM guidance, and knowledge graph, by P4GCN to enhance recommendation systems in our future works.

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

# A  DERIVATIONS

## A.1  The derivation of the upper bounds of $\ell_2$ sensitivity

We denote the adjacent databases by $\mathbf{A}$ and $\mathbf{A}'$ where $\mathbf{A}'_{km} = 1 - A_{km}$. And other elements of the two matrix are the same. The $k$th row in the $\tilde{\mathbf{L}}_{sym}$ of $\mathbf{A}$ is $\mathbf{l}_k$ (e.g., $\mathbf{l}'_k$ for $\mathbf{A}'$). Letting $d_j = \sqrt{\|\mathbf{a}_j\|_1}$, then we have

$$\|\mathbf{l}'_k - \mathbf{l}_k\|_2^2 = \left\| \left[ \frac{a_{kj}}{d_k d_j}, \cdots, \frac{a_{km}}{d_k d_m}, \cdots \right] - \left[ \frac{a_{kj}}{d'_k d_j}, \cdots, \frac{1-a_{km}}{d'_k d_m}, \cdots \right] \right\|_2^2$$

$$= \left\| \frac{d_k - d'_k}{d_k d'_k} \left[ \frac{a_{kj}}{d_j}, \cdots, \frac{1-a_{km}}{d_m}\frac{d_k}{d_k - d'_k} - \frac{a_{km}}{d_m}\frac{d'_k}{d_k - d'_k}, \cdots \right] \right\|_2^2$$

$$= (\frac{d_k - d'_k}{d_k d'_k})^2 \left( \sum_{j=1}^{N} \frac{a_{kj}^2}{\|\mathbf{a}_j\|_1} - \frac{a_{km}^2}{d_m^2} + \frac{\left( (1-a_{km})d_k - a_{km}d'_k \right)^2}{d_m^2(d_k - d'_k)^2} \right)$$

$$= (\frac{d_k - d'_k}{d_k d'_k})^2 \left( \sum_{j=1}^{N} \frac{a_{kj}}{\|\mathbf{a}_j\|_1} + \frac{\left( (1-a_{km})d_k - a_{km}d'_k \right)^2 - a_{km}^2(d_k - d'_k)^2}{\|\mathbf{a}_m\|_1(d_k - d'_k)^2} \right)$$

$$= (\frac{d_k - d'_k}{d_k d'_k})^2 \left( \sum_{j=1}^{N} \frac{a_{kj}}{\|\mathbf{a}_j\|_1} + \frac{\left( \frac{(1-a_{km})d_k - a_{km}d'_k}{d_k - d'_k} \right)^2 - a_{km}^2}{\|\mathbf{a}_m\|_1} \right)$$

$$\leq (\frac{d_k - d'_k}{d_k d'_k})^2 (\|\mathbf{a}_k\|_1 + \frac{\|\mathbf{a}_k\|_1 + a_{km}(1 - 2a_{km})}{\|\mathbf{a}_m\|_1})$$

$$\leq \frac{1}{\|\mathbf{a}_k\|_1^2 + \|\mathbf{a}_k\|_1} c_k + \frac{1}{\|\mathbf{a}_k\|_1} c_o \tag{11}$$

where $c_k = \sum_{j=1}^{N} \frac{a_{kj}}{\|\mathbf{a}_j\|_1} \leq \|a_k\|_1, c_o = \max_m \frac{1}{\|\mathbf{a}_m\|_1 + 1} \leq 1$. Then, we can obtain Eq.4.3.2 by replacing $d_k$ with its definition.

## A.2  Proof of Theorem 4.1

THEOREM A.1. *Given* $\mathbf{J} = \mathbf{LMN}$ *where all matrices are not zero matrices, there exists infinite combinations of* $\mathbf{N}' \neq \mathbf{N}, \mathbf{L}' \neq \mathbf{L}$ *such that* $\mathbf{J} = \mathbf{L}'\mathbf{MN}'$.

PROOF. Given $\mathbf{J} = \mathbf{LMN}, \mathbf{L} \in \mathbb{R}^{p \times q}, \mathbf{M} \in \mathbb{R}^{q \times r}, \mathbf{N} \in \mathbb{R}^{r \times s}$, we have

$$rank(\mathbf{LM}) = rank([\mathbf{LM};\mathbf{J}]) \tag{12}$$

Now we consider the equation

$$(\mathbf{L}'\mathbf{M})\mathbf{X} = \mathbf{J}, \mathbf{X} \in \mathbb{R}^{r \times s}, \mathbf{L} \neq \mathbf{L}' \tag{13}$$

As long as equation (13) is solvable, then we can directly set $\mathbf{N}'$ to be the solver $\mathbf{X}$, leading to the establishment of $\mathbf{J} = \mathbf{L}'\mathbf{MN}'$. Therefore, to make the equation (13) solvable, we must establish the following equation

$$rank(\mathbf{L}'\mathbf{M}) = rank([\mathbf{L}'\mathbf{M};\mathbf{J}])$$

Without loss of generality, we denote $\mathbf{L}' = \mathbf{L} + \Delta\mathbf{L}$. We now introduce a way to choose $\mathbf{L}'$ without changing $rank([\mathbf{L}'\mathbf{M}])$.

$$\mathbf{L}'\mathbf{M} = \mathbf{LM} + \Delta\mathbf{LM} \tag{14}$$

**Table 5: Parameters of layers in P4GCN**

| LayerName | Parameter |
|---|---|
| Local Agg. Weight | - |
| Social Agg. Weight | $\mathbf{W}_1 \in \mathbb{R}^{d \times d}$ and $\mathbf{W}_2 \in \mathbb{R}^{d \times d}$ |
| Fusion Layer | $\mathbf{W}_{fusion1} \in \mathbb{R}^{2d \times 2d}, \mathbf{W}_{fusion2} \in \mathbb{R}^{2d \times 2d}$ |
| Decoder | - |

By setting $\Delta\mathbf{L}$ as

$$\Delta\mathbf{L} = \begin{bmatrix} \delta_{11} & \cdots & 0 \\ \vdots & \ddots & \vdots \\ 0 & \cdots & 0 \end{bmatrix} \tag{15}$$

we can obtain that

$$\mathbf{L}'\mathbf{M} = \mathbf{LM} + \begin{bmatrix} \delta_{11}\mathbf{m}_{\cdot 1} \\ \vdots \\ 0 \end{bmatrix} = \mathbf{Z} + \Delta\mathbf{Z} = \begin{bmatrix} \mathbf{L}_1.\mathbf{M} + \delta_{11}\mathbf{M}_1. \\ \vdots \\ \mathbf{L}_p.\mathbf{M} \end{bmatrix} = \mathbf{Z}' \tag{16}$$

□

The influence of $\Delta\mathbf{Z}$ on the rank can be easily eliminated by setting a small enough value of $\delta_{11}$. In this way, the rank of $\mathbf{Z} = \mathbf{LM}$ is preserved as

$$rank(\mathbf{LM}) = rank(\mathbf{L}'\mathbf{M}) = rank([\mathbf{L}'\mathbf{M};\mathbf{J}]) \tag{17}$$

from which we can immediately infer that there exists at least a solver $\mathbf{X}$ such that $\mathbf{L}'\mathbf{MX} = \mathbf{J}$. Note that the choice of the position of value changing is not necessary to be specified to $(1, 1)$ and the number of changes is also not limited, there will thus be an infinite number of $\Delta\mathbf{L}$ that can be the alternative one, leading to the infinite number of combinations of $\mathbf{L}', \mathbf{N}'$. The distance between $\mathbf{L}'$ and $\mathbf{L}$ can be arbitrarily decided by choosing $\mathbf{L}' \leftarrow r\mathbf{L}', \mathbf{N}' \leftarrow \frac{1}{r}\mathbf{N}', r \in \mathbb{R}$ and $r \neq 0$

# B  THE ARCHITECTURE OF P4GCN

The architecture of P4GCN is shown in Table 5. During each iteration, the party $\mathcal{P}_1$ first inputs the batch data (e.g. the batched users' features $\mathbf{X}_{user,B}^{(0)}$ and the items' features $\mathbf{X}_{item}^{(0)}$) and the user-item graph into the local aggregation GC layer to obtain $\mathbf{X}_{user,B}^{(1)}$ and $\mathbf{X}_{item}^{(1)}$. Then, $\mathcal{P}_1$ uses sandwich encryption to make the social aggregation on users' features with $\mathcal{P}_2$ to obtain $\mathbf{X}_{user,B}^{(2)}$. $\mathcal{P}_1$ further fuses the two types of users' embeddings together by the fusion layer. Concretely, for each user $u_i$ in the current batch, its fusion of embeddings is $\mathbf{x}_{u_i}^{(3)} = [\mathbf{x}_{user,u_i}^{(1)\top}|\mathbf{x}_{user,u_i}^{(2)\top}]^\top \odot (\mathbf{W}_{fusion,u_i}[\mathbf{x}_{user,u_i}^{(1)\top}|\mathbf{x}_{user,u_i}^{(2)\top}]^\top) \in \mathbb{R}^{2d}$. Finally, both the items' embeddings $\mathbf{X}_{item}^{(1)}$ and the users' embeddings $\mathbf{X}_{user,B}^{(3)} = [\mathbf{x}_{u_1}^{(3)}, ..., \mathbf{x}_{u_B}^{(3)}]$ will be input into the decoder to predict the rating $\hat{r}_{u,v} = 4*sigmoid(Relu\left([\mathbf{x}_{user,u}^{(3)\top}|\mathbf{x}_{item,v}^{(1)}]\mathbf{W}_{mlp1}\right)\mathbf{W}_{mlp2})$.

The LightGCN used in our experiments shares the same architecture as our P4GCN but without the fusion layer. We directly cat the users' embeddings $\mathbf{X}_{user,B}^{(1)}$ and the items' embeddings $\mathbf{X}_{item}^{(1)}$

together and then input them into the two-layer decoder to obtain the prediction.

## C HOMOMORPHIC ENCRYPTION

### C.1 Paillier algorithm

Paillier is a public-key cryptosystem that supports additive homomorphism [39]. The main steps of the Paillier algorithm are key generation, encryption, and decryption.

*Key generation.* First randomly selects two large prime numbers $p$ and $q$ that satisfy the formula $gcd(pq, (p-1)(q-1)) = 1$, and $p$, $q$ are equal in length. Then we calculate $n = pq$ and $\lambda = lcm(p-1, q-1)$. Second, randomly selection of integer $g \in Z_{n^2}^*$ and define function $L$ as $L(x) = \frac{x-1}{n}$ and calculate $\mu = \left( L \left( g^\lambda \bmod n^2 \right) \right)^{-1} \bmod n$. Finally, we get private key $(n, g)$ and public key $(\lambda, \mu)$.

*Encryption.* First input the plaintext $m$ satisfies $0 \leq m \leq n$. Then choose a random number $r$ that satisfies $r \in Z_n^*$. Finally, we calculate the ciphertext as $c = g^m r^n \bmod n^2$.

*Decryption.* Input ciphertext $c$ that satisfies $c \in Z_{n^2}^*$, and then calculate the plaintext message as $m = L \left( c^\lambda \bmod n^2 \right) \cdot \mu \bmod n$

## D LIMITATION AND BROADER IMPACT

This work introduces a way to leverage user's social data to improve the recommendation system on the company view. One limitation lies in that we only discuss the method on GCN operator. And we plan to extend this work to other operators like graph attention as our future work.

