# OpenReview forum: "P4GCN: Vertical Federated Social Recommendation with Privacy-Preserving Two-Party Graph Convolution Network"
_ACM.org/TheWebConf/2025/Conference — WWW 2025 Poster_

### Official Review · Reviewer_uP2G · 2024-11-23

**Novelty:** 3
**Technical Quality:** 3

**Review:**

The paper proposes a sandwich encryption module to ensure the data privacy of GNN training data. The quality of the paper is average, and there are some logical problems: for example, in figure 1, there are two types of graphs, user-user graph and user-item graph. The graph neural network in equation (1)-(3) only uses the adjacency matrix of the user-user graph and the user features in the user-item graph. However, the adjacency matrix of the user-item graph do not seem to be fully utilized, and these features still seem to be important.

**Questions:**

1.It is recommended that the authors include a concise summary at the end of the method section. Specifically, providing the complete formula for the entire P4GCN layer would greatly enhance the clarity of the calculation process for both model input and output. Additionally, the updated formulas for W_{\mathcal{P}_1} and W_{\mathcal{P}_2} during backpropagation should be fully written in terms of Z, X and \mathcal{L}, so that the reader can better follow the model’s training process.
2.P4GCN seems to have limitations in capturing the relationship between the same user and different items in user-item interactions. Could the authors provide a more detailed explanation of how the model learns these relationships, particularly in the context of Figure 3? A clearer description would help readers understand how interactions between a user and all the items they engage with are modeled.
3.The method used for comparative experiments is relatively outdated, and the author should add relevant baselines from the past two years for comparison.
4.The design of the decoder in the proposed method appears to play a crucial role in the model's performance. However, the paper does not explain what the decoder is, nor does it justify why such a specific structured decoder is used. A discussion on this would help readers understand its significance and contribution to the model’s success.

**Reviewer Confidence:**

3: The reviewer is confident but not certain that the evaluation is correct

**Scope:**

3: The work is somewhat relevant to the Web and to the track, and is of narrow interest to a sub-community

---

### Official Review · Reviewer_5cWb · 2024-11-27

**Novelty:** 5
**Technical Quality:** 5

**Review:**

This paper focuses on privacy-preserving models for social graph recommendation tasks. Specifically, a P4GCN model is proposed to jointly optimize model parameters with private data from two parties without causing information leakage. A sandwich encryption solution is used to ensure information protection during the training and inferencing of the GCN model. Theoretical analyses guarantee the privacy protection, and experimental results varify the model performance.

Pros.
+ A novel federated social recommendation framework that considers two-party GCNs.
+ Experimental results verify the superiority of P4GCN.
+ The paper is well written and easy to follow.

Cons.
- The local baselines are mostly classical methods or be without social data, where models like NeuMF and LightGCN have shown competitive performance among them. It would be more persuasive to compare with more powerful methods with accessible social data as a strong baseline to reflect the influence of data privacy requirements.
- The overall architecture is restricted to graph convolution-based models.
- The DP characteristics of the overall model requires further explanation.

**Questions:**

1. From the general comparison, it seems that P4GCN with privacy budget $\epsilon$ even outperforms ideal cases. Could you explain why?
2. The $\epsilon$-DP of aMGM has been demonstrated already. Could you prove the same holds true for the general P4GCN model?

**Reviewer Confidence:**

2: The reviewer is willing to defend the evaluation, but it is likely that the reviewer did not understand parts of the paper

**Scope:**

4: The work is relevant to the Web and to the track, and is of broad interest to the community

---

### Official Review · Reviewer_qn1W · 2024-11-30

**Novelty:** 5
**Technical Quality:** 3

**Review:**

**Overview**

This paper proposes a novel vertical federated social recommendation method called P4GCN, which introduces a sandwich encryption technique to protect user-side social networks. This approach separates the social domain and the interaction domain into two independent parts, allowing the interaction domain to utilize any graph recommendation method to obtain user and item embeddings. Experiments conducted on four social recommendation datasets demonstrate the effectiveness of the proposed method.

---

**Pros**
1. The paper is well-structured, and the problem under investigation holds practical significance.
2. The idea of using encryption methods to separate the interaction domain and the social domain is quite intriguing.
3. Extensive analytical experiments demonstrate the effectiveness of the proposed method from various perspectives.

**Cons**
1. The motivation of the paper is not sufficiently well-developed. In the scenario outlined in Figure 1 of the paper, where Company A retains the social network among users, it raises the question: why can't a complete GCN operation be directly performed on Company A's side, followed by the application of simple Laplacian noise?
2. The sandwich encryption employed in the paper incorporates both homomorphic encryption and differential privacy. However, the authors did not provide a detailed analysis of the computational overhead associated with this approach, raising concerns about the practical applicability of the proposed method.
3. In the experimental section, the paper only compares traditional GCN-based methods (e.g., LightGCN), but it does not include comparisons with social recommendation methods that utilize GCNs (e.g., DiffNet, DESIGN, SEPT, etc.).

**Questions:**

The authors address the social recommendation problem in a vertical federated scenario. However, in horizontal federated scenarios, the problem becomes significantly more complex. For example, the server cannot access any user's private data, making it almost impossible to perform global aggregation using GCN. It would be interesting to see the authors' perspective on this issue and whether they could analyze the similarities and differences between the proposed method and traditional horizontal federated recommendation methods (e.g., FedMF, FedGNN).

**Reviewer Confidence:**

3: The reviewer is confident but not certain that the evaluation is correct

**Scope:**

4: The work is relevant to the Web and to the track, and is of broad interest to the community

---

### Official Review · Reviewer_P9zZ · 2024-12-02

**Novelty:** 3
**Technical Quality:** 3

**Review:**

This paper studies Privacy-Preserving based social recommendation, which aims to improve the social recommendation system without direct access to social data. To this end, the authors propose the Sandwich-Encryption module that ensures data privacy throughout the collaborative computing process and provides a theoretical analysis of the security guarantees.

Strong points:
1.	the authors study a useful problem for social recommendation
2.	the proposed method is feasible and practical.

Weak points:
1.	The problem formulation lacks clarity. Key terms, such as “party” and “social-data-fully-inaccessible”, are not adequately explained or formally defined. The authors need to emphasize and clearly define the target of the problem.
2.	Figure 2 requires improvement. Specifically:
a)	The input and output of the framework are not clearly depicted.
b)	The symbols in the figure are not explained, making it difficult to intuitively understand the processing flow of the framework.
3.	The paper includes a large number of symbols in various forms, which complicates comprehension. The authors should simplify the symbolic system or provide a comprehensive notation table to facilitate understanding and reference.
4.	Neither the dataset nor the source code is provided, which limits reproducibility and validation.
5.	The methods used for comparison are all from 2022 or earlier. The authors should include comparisons with more recent methods from the past two years.

**Questions:**

1. The proposed framework is designed based on GCN. Can this framework be applied to other GNN architectures, such as GAT?
2. see the weak points

**Reviewer Confidence:**

2: The reviewer is willing to defend the evaluation, but it is likely that the reviewer did not understand parts of the paper

**Scope:**

3: The work is somewhat relevant to the Web and to the track, and is of narrow interest to a sub-community

---

### Official Review · Reviewer_Dd76 · 2024-12-02

**Novelty:** 5
**Technical Quality:** 5

**Review:**

This research proposes a new algorithm to privately learn GNN models for solving recommendation problems where one party holds user-item interaction data and another party holds social user-user interaction data. The newly proposed algorithm P4GCN identifies that both the forward pass and backward passes (one for weight matrix and one for node features) of their GCN network can be expressed in the form J = LMN where L and N are held by one party and J and M are held by the other party. The algorithm utilizes 2 techniques to accomplish this calculation privately. (1) It uses homomorphic encryption to share M to compute an encrypted J that can only be decoded by the party that holds M but computed by the other party. (2) It uses DP noise on J such that the value of J and M cannot leak information about L and N. This technique is called sandwich encryption. The authors conduct a series of experiments on 3 datasets: Filmtrust, CiaoDVD, Douban and Epinions and compare against 2 different type of baseline methods: (1) using only user-item interaction (2) MF based methods using both user-item and user-user interactions. Against both baselines, P4GCN has favorable results as it outperforms most of them on both RMSE and MAE metrics. The authors also conduct experiments to understand the impact of DP noise on the algorithm as well as ablation studies to understand the impact of both the fusion layer as well as experimenting with different base architectures (PMF, NeuMF, GCN)

Pros:

- Using GCNs for recommendation tasks which use both user-user data and user-item data seems like an important direction for performance but privacy reasons might prevent this from happening. This paper does a good job of showing that we can do this privately with good results.
- The sandwich encryption method could potentially be applicable in other scenarios as well.


Cons:
- Its not clear how statistically significant the results are. Can the authors provide some kind of variance or confidence intervals? Are the experiments performed multiple times or only once?

**Questions:**

For setting $\beta$, does it depend on your privacy budget? For example, in Figure 5, it seems like the value of $\beta$ is larger in the non-private setting as expected on some datasets but not others. Can the authors give some intuition on why some of the results here are not as expected?

**Reviewer Confidence:**

3: The reviewer is confident but not certain that the evaluation is correct

**Scope:**

3: The work is somewhat relevant to the Web and to the track, and is of narrow interest to a sub-community